# Glycoscience in Advancing PD-1/PD-L1-Axis-Targeted Tumor Immunotherapy

**DOI:** 10.3390/ijms26031238

**Published:** 2025-01-31

**Authors:** Qiyue Sun, Senlian Hong

**Affiliations:** State Key Laboratory of Natural and Biomimetic Drugs, Chemical Biology Center, Department of Chemical Biology, School of Pharmaceutical Sciences, Peking University, Beijing 100191, China; cherry99bu88@163.com

**Keywords:** glycosylation, PD-1/PD-L1 blockades, immunotherapy

## Abstract

Immune checkpoint blockade therapy, represented by anti-PD-1/PD-L1 monoclonal antibodies, has significantly changed the immunotherapy landscape. However, the treatment is still limited by unsatisfactory response rates, immune-related adverse effects, and drug resistance. Current studies have established that glycosylation, a common post-translational modification, is crucial in promoting cancer progression and immune invasion. Targeting aberrant glycosylation in cancers presents precision medicine regimens for monitoring cancer progression and developing personalized medicine. Notably, the immune checkpoints PD-1 and PD-L1 are highly glycosylated, which affects PD-1/PD-L1 interaction and the binding of anti-PD-1/PD-L1 monoclonal antibodies. Recent achievements in glycoscience to enhance patient outcomes, referred to as glycotherapy, have underscored their high potency in advancing PD-1/PD-L1 blockade therapies, i.e., glycoengineered antibodies with improved binding toward PD-1/PD-L1, pharmaceutic inhibitors for core fucosylation and sialylation, and synergistic treatment with the antibody–sialidase conjugate. This review briefly introduces the PD-1/PD-L1 axis and glycosylation and highlights the fundamental and applied advances in glycoscience that improve PD-1/PD-L1 immunoblockade therapies.

## 1. Introduction

Immunotherapy mobilizes the patient’s immune system to destroy tumors. For example, immune checkpoint inhibitors (ICIs), exemplified by programmed cell death protein 1/ programmed cell death 1 ligand 1 (PD-1/PD-L1) inhibitors, result in durable cancer remission and have revolutionized cancer treatment (Figure 1). Nevertheless, drawbacks of PD-1/PD-L1 blockade therapies are beginning to emerge. Single-agent response rates of anti-PD-1/PD-L1 antibodies remain at around 15%–30%. Concurrently, heterogeneous responses have been seen between distinct tumors in the same patient. In addition, patients with breast, prostate, and colon cancers show a low frequency of response, which significantly reduces the clinical benefit [1]. In addition, immune checkpoint blockade (ICB) therapies are associated with various immune-related adverse events (e.g., skin toxicity) [2]. Thus, developing new therapeutic strategies to improve the efficacy and safety of PD-1/PD-L1 blockade therapies is imperative.

Extensive studies have shown that most immune checkpoints, including PD-1 and PD-L1, are glycoproteins [3]. Among the post-translational modifications (PTMs), glycosylation is one of the most prevalent and diverse forms, with more than half of human proteins being glycosylated [4]. In addition, alterations in tumor-associated glycans affect many biological processes that control tumor pathogenesis and drug resistance, making glycosylation a preferred drug target [5,6,7]. Glycans also present promising biomarkers for evaluating and predicting the effectiveness of immunotherapy in patients. Therefore, this article focuses on the role of glycosylation in tumor biology and immunotherapy.

## 2. PD-1/PD-L1 Axis

Programmed cell death protein 1 (PD-1) is predominantly expressed in activated T cells, B cells, dendritic cells, monocytes, and natural killer cells [8,9]. The intracellular domain of PD-1 contains an immunoreceptor tyrosine-based inhibitory motif (ITIM) and an immunoreceptor tyrosine-based switch motif (ITSM) [10,11]. The extracellular IgV-like domain of PD-1 recognizes two ligands: programmed cell death 1 ligand 1 (PD-L1, also known as B7-H1 or CD274) and programmed cell death 1 ligand 2 (PD-L2, also known as B7-DC or CD273) [9]. PD-L1 is usually expressed on antigen-presenting cells (APCs) and tumor cells. When PD-1 interacts with PD-L1, the tyrosine residues of ITIM and ITSM on PD-1 are phosphorylated by kinases (Lck and/or Src kinases) in T cells, resulting in the recruitment of Src homology-2 (SH2)-domain-containing tyrosine phosphatase-2 (SHP-2) [12,13]. SHP-2 then dephosphorylates several key downstream kinases, inhibiting T cell biological functions and CD28 signaling [10,14]; suppressing the immune response; and promoting tumor cell epithelialization, metastasis, and infiltration [15,16,17,18].

Monoclonal antibodies (mAbs) can block the PD-1/PD-L1 inhibitory pathway, reactivate the recognition and cytotoxicity of immune cells, and avoid the immune escape of tumor cells [19,20]. Nowadays, PD-1/PD-L1 blockades are popular, alone or combined with chemotherapy, to improve patient outcomes of many solid tumors in different settings (neoadjuvant, adjuvant, and metastatic), as summarized in Table 1.

## 3. Glycosylation

Glycosylation involves an enzymatic network by which saccharides form glycosidic bonds with other saccharides, proteins, lipids, or RNAs [21,22]. The resulting glycoconjugates are primarily classified by the nature and linkage of their non-glycosyl components [23]. For instance, glycoproteins are a type of glycoconjugate that carry one or more glycans. Protein glycosylation is classified briefly by linkages as *O*-glycosylation, *N*-glycosylation, *C*-glycosylation, and GPI-anchoring [23,24]. Of these, *N*-glycosylation and *O*-glycosylation are the most common types. The glycosylation of proteins begins in the endoplasmic reticulum (ER) lumen by glycosyltransferases (GTs). Subsequently, the protein enters the Golgi apparatus, where glycosidases perform carbohydrate trimming for the final attachment of other carbohydrate residues to produce matured glycoproteins [25]. In cancer cells, glycosylation changes are broadly detected, including hypersialylation, extensive core fucosylation, *O*-glycan truncation, and *N*- and *O*-glycan hyperbranching [21].

Research on how glycosylation affects the function of cancer proteins can be divided into two categories: altered glycosylation of cell surface receptors that promotes cancer growth or prevents cancer growth [26]. A common feature of cancer cells is the increasing dependence on glucose metabolism, a metabolic reconfiguration known as the Warburg effect that supports the high energy demand and biosynthesis [27,28]. Subsequently, the flux of the hexosamine biosynthesis pathway (HBP) increases, leading to the production of uridine diphosphate-*N*-acetylglucosamine (UDP-Glc*N*Ac), a key metabolite involved in *N*- or *O*-glycosylation [26,29]. This changes the synthesis of glycoconjugates in cancer cells [30]. Also, many studies have elucidated the impact of glycosylation on tumor progressions, such as controlling tumor metastasis and immune invasions. In addition, recent studies have demonstrated that the PTMs of PD-L1 control its stability and interaction with PD-1 [31]. Accumulating evidence has shown that PD-1/PD-L1 is highly glycosylated [32,33]. This section will focus on the changes in cancer *N*-glycosylation and *O*-glycosylation and discuss how glycosylation affects tumor biology and anti-tumor immunity in the tumor microenvironment (TME).

### 3.1. N-glycosylation and O-glycosylation

#### 3.1.1. *N*-glycosylation

During *N*-glycosylation, precursor oligosaccharides are attached to the asparagine (Asn, N) side-chain nitrogen in the consensus sequence Asn-X-Ser/Thr, where X is any amino acid except proline. Mature *N*-glycans are produced in the ER and Golgi apparatus via sequential glycosylation [34,35,36,37]. *N*-glycosylation broadly affects tumor progression, including aberrant cell–matrix interactions, disruption of signaling pathways, enhanced invasiveness, and immune evasion [23,26,35,38]. Notably, increased *N*-glycan branching modulates tumor cell–stroma interactions and promotes tumor cell migration, regarded as a hallmark of cancers [21].

Additionally, glycosylated protein typically presents a heterogeneous pattern on Western blots. PD-L1 shows a series of bands of approximately 45 kDa. Treated by the recombinant glycosidase peptide-N-glycosidase F (PNGase F), the *N*-glycans can be effectively removed, resulting in a single PD-L1 band of 33 kDa. This indicates that PD-L1 mainly undergoes *N*-glycosylation [32]. Further analysis has revealed that four asparagine residues (N35, N192, N200, and N219) in the PD-L1 extracellular domain are *N*-glycosylation sites [32]. Additionally, PD-1 N192, N200, and N219 glycosylation can impede the binding of glycogen synthase kinase 3β (GSK3β) and PD-L1, thereby preventing subsequent proteasomal degradation [32]. These modifications eventually promote PD-L1/PD-1 interaction [32,33]. Additionally, it has been proved that *N*-glycosylation is crucial to the cell surface localization of these proteins. Furthermore, PD-1 in tumor-specific T cells is broadly *N*-glycosylated [3]. Four potential *N*-glycosylation sites (N49, N58, N74, and N116) have been identified in the IgV domain of PD-1 [39]. Furthermore, the glycosylation of PD-1 N58 can enhance the interaction of mAbs [40,41]. These findings indicate that *N*-glycosylation of PD-1/PD-L1 matters in regulating PD-1/PD-L1 expression and stability. Further study of this complex process may lead to breakthroughs in tumor immunotherapy.

#### 3.1.2. *O*-glycosylation

During *O*-glycosylation, glycans are attached to the hydroxyl groups of serine (Ser), threonine (Thr), or hydroxylysine (Hyl) in the peptide chain, forming an *O*-glycosidic bond [15,35]. In contrast to *N*-glycosylation, which typically occurs within the ER and Golgi apparatus, *O*-glycosylation is predominantly observed in the nucleus and cytoplasm. The two most prevalent *O*-glycosylation types are *O*-acetylgalactosamine (*O*-Gal*N*Ac) and *O*-acetylglucosamine (*O*-Glc*N*Ac) [23]. *O*-Glc*N*Acylation regulates nutrient sensing, metabolism, signal transduction, and transcriptional processes and plays a significant role in developmental processes and physiological and pathological processes [42]. Studies have demonstrated that *O*-Glc*N*Acylation targets the endosomal sorting complexes required for the transport-mediated intracellular sorting pathway of PD-L1 proteins, ultimately blocking PD-L1 lysosomal degradation and promoting tumor immune evasion [6]. *O*-glycosyltransferase (OGT) has also been demonstrated to influence tumor progression. Research indicates that elevated levels of OGT in exosomes derived from esophageal cancer stem cells (ECSCs) upregulate PD-1 in CD8^+^ T cells, thereby suppressing CD8^+^ T cells. This provides a novel mechanism of immune evasion [43]. Further study of *O*-glycosylation and its impact on the PD-1/PD-L1 axis would be beneficial.

### 3.2. Modifications That Can Extend the Structure of Glycoconjugates

Surface *N*-/O-glycan chains are modified by adding Glc*N*Ac, galactose, sialic acid, and fucose sugars [44]. In this section, we review fucosylation and sialylation, with a focus on the potential of targeting these modifications to promote PD-1/PD-L1 blockade therapies.

#### 3.2.1. Fucosylation

Fucosylation contributes to cancer progression. Fucosylated glycans are catalyzed by a series of fucosyltransferases (Fuc-Ts; Fuc-TI-Fuc-TXI, encoded by *FUT1*-*FUT11*, where *FUT3* is also known as the Lewis gene Le). Fucosylation is typically subdivided into two categories: terminal fucosylation, which generates specific Lewis blood group antigens, such as Le^x^ and Le^y^ or Le^a^ and Le^b^, and core fucosylation [21]. A study demonstrated that four *N*-glycosylation sites on PD-1, particularly N49 and N74, undergo broad core fucosylation. The related Fut8 is markedly upregulated in various cancer types. Inhibiting Fut8 by genetic ablation or pharmacological inhibition has been demonstrated to reduce the cell surface expression of PD-1 and enhance T cell activation, promoting effective tumor eradication [45].

#### 3.2.2. Sialylation

Sialylation means attaching sialic acid units to the termini of glycans [46]. Sialic acid, referring to nine-carbon saccharides that are present in all mammalian cells, is predominantly located at the non-reducing ends of *N*- and *O*-glycans [47,48]. In mammals, sialic acids mainly are *N*-acetylneuraminic acid (Neu5Ac) and its derivative or *N*-glycolylneuraminic acid (Neu5Gc) [47]. It has been demonstrated that Neu5Gc is upregulated in many cancers, making it an intriguing target for developing mAbs directly against tumor-associated glycans. In addition, Neu5Gc has been demonstrated to modulate glyco-immune checkpoints [48]. Notably, hypersialylation, particularly high levels of α2,6-sialosides, is linked to a range of cancers relating to metastatic phenotypes and poor prognosis [21,49]. Concurrently, elevated sialylation levels promote tumor cell proliferation [50]. Studies have demonstrated that the genetic and therapeutic desialylation of tumor cells delays tumor growth, rejuvenates tumor-infiltrating T cells, and leads to the anti-tumor polarization of tumor-associated macrophages [51]. In addition, Cornelissen et al. illustrated that glioblastoma cells induce a myeloid immune-suppressive phenotype, which can be partially reversed by trimming glioblastoma-associated sialic acids [52]. Furthermore, evidence suggests that sialyltransferases are associated with multiple cancers [47]. These studies support the tumor therapeutic potential of targeting sialylation.

Sialosides can interact with sialic-acid-binding immunoglobulin (Ig)-like lectins (Siglecs) that have immune-modulatory domains [53]. Siglecs function as inhibitory co-receptors and activating receptors for immune cell signaling in different contexts [54]. Upon binding of sialosides with inhibitory Siglecs, the intracellular domains of Siglecs can be phosphorylated by locally activated Src kinases, recruiting SHP-1 or SHP-2 phosphatases that decouple the downstream cellular activation [48]. Siglecs are expressed on tumor-infiltrating T cells, myeloid cells, and natural killer (NK) cells. The interaction between Siglecs and sialosides is a glyco-immune checkpoint for tumor cells. It has been demonstrated that tumor-infiltrating T cells upregulate multiple inhibitory Siglecs, including Siglec-3, Siglec-5, Siglec-7, Siglec-9, and Siglec-10 [54]. Siglecs bind to a range of sialoglycans, and their binding capacity can be influenced by the underlying glycan or additional modifications [48,54]. The hypersialylation in cancer tissues has been demonstrated to promote tumor immune evasion through the engagement of Siglecs [48]. Among these is Siglec-9, predominantly expressed in myeloid cells, that can bind to sialylated MUC1, subsequently upregulating PD-L1 [55].

## 4. Glycoscience in Advancing PD-1/PD-L1 Blockade Therapies

Targeting glycoconjugates may revolutionize cancer therapy. Here, we focus on glycoscience that improves PD-1/PD-L1 blockade therapies by offering new targets, optimizing mAbs, or serving as predictive biomarkers (Figure 2).

### 4.1. Targeting Glycan

The loss of core fucosylation markedly enhances PD-1 ubiquitination, resulting in PD-1 degradation within the proteasome [56]. Interrupting core fucosylation offers a promising strategy to improve anti-tumor immune responses [45,57]. In addition, specific glycan signatures found on tumor cells are regarded as a novel immune checkpoint [5]. Siglecs are representative glyco-immune checkpoints [54]. Siglec-7/Siglec-9 blockades can notably reduce the tumor burden in vivo, enhancing anti-tumor immunity [58]. Haas et al. demonstrated that targeting the tumor-restricted, glycosylation-dependent sialosides/Siglec-9 axis can release T cell subsets within the melanoma TME, while simultaneously limiting T cell activation to the TME, reducing the potential for uncontrolled T cell activation and immune-related adverse events [42]. Wang et al. found that Siglec-15 is frequently overexpressed on human cancer cells and tumor-infiltrating myeloid cells, and its expression is mutually exclusive with that of PD-L1. In some mouse models, the genetic ablation or antibody blockade of Siglec-15 has been demonstrated to enhance anti-tumor immunity in the TME [59]. He et al. prepared an anti-Siglec-15 mAb and demonstrated that it could moderately block tumor growth in vivo and in vitro [60]. These findings provide a compensational strategy for patients who are not responding to PD-1/PD-L1 therapies. These antibodies against Siglec-7, Siglec-9, and Siglec-15 enhance anti-tumor immunity and limit cancer progression, providing an option for patients with PD-L1 resistance [42,58,59,60,61,62] (Figure 2A).

Inhibitors targeting other glyco-immune checkpoints have been developed. Galectin-3 (Gal-3) is a β-galactoside-binding lectin highly expressed in the TME of invasive cancers. A preclinical study demonstrated that combining an orally active Gal-3 antagonist with the PD-L1 blockade inhibitor promotes tumor immune infiltration, reduces lung adenocarcinoma growth, and blocks metastasis [63]. Wang et al. proposed a new Gal-3 inhibitor that synergized with PD-L1 immune checkpoint inhibitors [63,64]. Combination therapy of inhibiting glyco-immune checkpoints and PD-1/PD-L1 blockades is promising. Clinical trials are underway to block the Gal-3/LAG3 or Gal-9/TIM3 axis in conjunction with anti-PD-1 therapy. In 2022, relatlimab (an anti-LAG-3 antibody) in combination with nivolumab was approved by the FDA and the European Medicines Agency (EMA) for treating adult and pediatric patients with unresectable or metastatic melanoma. Bispecific antibodies targeting PD-1 and LAG-3 have also been developed and are under clinical investigation in solid tumors [65]. The combination of anti-LAG-3 mAbs and pembrolizumab was also assessed in patients with metastatic NSCLC and head and neck cancer (NCT03625323). However, results from an open-label phase II study (NCT03662659) showed that in patients with gastric or gastroesophageal adenocarcinoma, the objective response rate (ORR) is reduced in patients receiving the combination regimen of relatlimab plus nivolumab and chemotherapy versus those receiving only nivolumab plus chemotherapy [66]. In addition to the ongoing trials, further clinical studies on TIM-3 inhibitors in combination with anti-PD-1/PD-L1 mAbs are needed.

### 4.2. Targeting Glycosylation

Targeting *N*-glycosylation on PD-L1/PD-1 represents a novel therapeutic strategy. D-mannose is the major monosaccharide residue of *N*-glycans. It has been demonstrated that D-mannose can activate AMP-activated protein kinase (AMPK) to phosphorylate PD-L1 at S195, leading to abnormal glycosylation and proteasomal degradation of PD-L1 (Figure 2B). The orally active D-mannose has been demonstrated to downregulate PD-L1 in mice and inhibits PD-L1/PD-1 interaction. Notably, the combination of D-mannose with the anti-PD-1/PD-L1 blockade exhibits a pronounced inhibitory effect on the growth of triple-negative breast cancer (TNBC) and extends the lifespan of tumor-bearing mice [67]. Similarly, studies have shown that AMPK activated by metformin directly phosphorylates S195 of PD-L1 [68]. In addition, the dietary polyphenol resveratrol (RSV) has been demonstrated to operate as a direct inhibitor of glyco-PD-L1-processing enzymes (α-glucosidase/α-mannosidase) to induce aberrant *N*-glycosylation of PD-L1. RSV has been shown to promote PD-L1 dimerization, impede the localization of PD-L1 to the plasma membrane, and ultimately enhance cytotoxic T lymphocyte immune surveillance against tumor cells. This provides experimental evidence to support the combination of RSV with ICIs [69]. However, the oral bioavailability of RSV is considerably less than 1% [70]. Moreover, the interaction between RSV and PD-L1 is complex [69]. These may restrict the applications of RSV. In addition, Zhang et al. found that niclosamide, an inhibitor acting on both HuR translocation and PD-L1 glycosylation, exhibits potent cytotoxicity against MDA-MB-231 with a half-maximal inhibitory concentration (IC_50_) of 1.07 μM, showing potential to impair immunotherapy resistance [71].

Additionally, *O*-glycosylation in tumor cells regulates the interactions of tumor cells with macrophages/cytotoxic T lymphocytes. Itraconazole, a Gal*N*Ac-type *O*-glycosylation inhibitor, has shown to reshape the TME. Using itraconazole and anti-PD-1 mAbs effectively inhibits tumor growth in vivo [72]. Targeting altered PTMs to normalize metabolic flux may also provide a viable strategy. One study demonstrated that blocking T255 O-GlcNAcylation of phosphoglycerate kinase 1 decreases colon cancer cell proliferation, suppresses glycolysis, enhances the TCA cycle, and inhibits tumor growth in xenograft models [73].

### 4.3. Targeting Glycoside Hydrolases and Glycosyltransferases (GTs)

Targeting GTs and glycosidases represents a promising therapeutic avenue. Combining targeting glycosyltransferases with anti-PD-1/PD-L1 mAbs may optimize PD-1/PD-L1 blockade therapies. A study demonstrated that OSMI-4 (an *O*-Glc*N*Ac transferase inhibitor) promotes the transport of intracellular PD-L1 from early endosomes to lysosomes for degradation (Figure 2B). Compared to monotherapy, combining OSMI-4 with the anti-PD-L1 mAb presents a synergistic inhibitory effect on the growth of hepatocellular carcinoma (HCC) and melanoma [6]. Xu et al. postulated that fucosyltransferase VII (FUT7) remodels the glucose metabolism and TME of TNBC, becoming a potential target for enhancing ICB treatment [74].

GTs affect broad physiological processes. Consequently, directly targeting GT may result in adverse side effects. Reported OGT inhibitors include substrate analogs, dual-substrate inhibitors, and high-throughput screening inhibitors [75]. These OGT inhibitors have some drawbacks in terms of their clinical applications, as shown by the reducing HBP flux and the jeopardized glycan synthesis, poor cell permeability, poor selectivity, and off-target toxicity [76]. In comparison to GT inhibitors, designing small molecular inhibitors to competitively inhibit the glycosylation of target proteins without interfering with the overall cellular glycosylation process may represent a promising alternative approach. Zhu et al. chemically synthesized CPP-G1 peptide containing hepatocyte growth-factor-regulated tyrosine kinase substrate (HGS) glycosylation sites. Treating SK-Hep-1 cells with different concentrations of CPP-G1 peptide decreased the expression of HGS *O*-Glc*N*Acylation and PD-L1 in a dose-dependent manner but had no significant effect on the cellular *O*-Glc*N*Acylation level. The authors demonstrated that using short peptides to competitively inhibit HGS glycosylation can specifically target intracellular protein *O*-Glc*N*Acylation, regulating the lysosomal degradation of PD-L1 [6].

### 4.4. Optimizing PD-1/PD-L1-Blocking Therapeutic Antibodies

#### 4.4.1. Monoclonal Antibodies Targeting Glycosylated PD-L1 (gPD-L1) and PD-1 (gPD-1)

In a recent study, Li et al. demonstrated that STM108, a glycosylation-specific mAb targeting gPD-L1, induces gPD-L1 internalization, effectively inhibits the PD-1/PD-L1 axis, and enhances anti-tumor immunity in mice. Additionally, an antibody–drug conjugate comprising STM108 and the potent anti-mitotic drug monomethyl auristatin E was generated. It led to superior survival outcomes compared to mice treated with the STM108 control. Furthermore, it has been demonstrated that this approach also induces effective cell killing and bystander killing of neighboring cancer cells that lack PD-L1 expression. Moreover, it was demonstrated that mAbs that recognize the gPD-L1 N192/N200 site, rather than those that specifically target N35, are the only mAbs that can exert induction effects. As normal tissues and primary immune cells express minimal levels of PD-L1 and gPD-L1, the majority of cells targeted by gPD-L1-ADC are located within the tumor area. This reduces off-target toxicity, resulting in safe clinical application [33]. This work demonstrated the potential of targeting glycosylated protein to improve ICBs and antibody–drug conjugates. Sun et al. screened the mAb STM418, which specifically targets gPD-1. In comparison with FDA-approved anti-PD-1 antibodies, it displayed a higher binding affinity for gPD-1, effectively inhibited the PD-L1/PD-1 interaction, and enhanced anti-tumor immunity [3]. In addition, Wang et al. developed MW11-h317 (an mAb-targeting gPD-1). The crystal structure revealed that the N58 glycosylation of PD-1 is a critical determinant in the binding process. The IC_50_ values of MW11-h317 and nivolumab on PD-1/PD-L1 binding were 1.4 and 1.3 nM, respectively. This indicated that the inhibitory effect of MW11-h317 on PD-1/ligand binding is similar to that of nivolumab [77]. Liu et al. conducted an interface analysis on the X-ray crystal structure of the fully human mAb mAb059c fragment antigen binding (Fab) complexed with the PD-1 extracellular domain. Similarly, their findings revealed that N58 in the BC loop plays a crucial role in mediating the interaction between mAb059c and PD-1. They also found that mAb059c blocks PD-1 and PD-L1 interaction, with an IC_50_ of 1.6 nM [78]. These studies collectively suggest that targeting gPD-1/PD-L1 may represent a promising avenue for enhancing immune checkpoint therapy.

#### 4.4.2. Antibody–Drug Conjugates (ADCs)

ADCs represent one of the most rapidly expanding areas of biopharmaceutical research and development (Figure 2C). They are produced by linking cancer-cell-specific mAbs to highly potent cytotoxic drugs [79]. However, the toxicity, drug resistance, and intra- and inter-tumor heterogeneity of ADCs limit their clinical application. In response, researchers have improved the ADC components (antibodies, linkers, payloads, and conjugation chemistry), for example, by developing bispecific ADCs, probody–drug conjugates (PDCs), immune-stimulating antibody conjugates (ISACs), protein-degrading antibody conjugates (DACs), and dual-payload ADCs [80]. Notably, random conjugation often results in heterogeneous mixtures of ADCs, ultimately leading to less efficient payload delivery. Therefore, there is a need to develop site-specific conjugation methods with moderate stability. Oligosaccharide-based conjugation offers distinctive advantages. The payload can be conjugated to the oligosaccharide N297 site in the CH_2_ domain, circumventing the need to conjugate through amino acid residues [79]. It has also been shown that attaching small-molecule payloads to the CH_2_ domain can compensate for the instability of glycosylation-free antibodies [81]. Zhou et al. developed an artificial anti-PD-L1 antibody–sialidase conjugate that specifically recognizes gPD-L1 on the tumor surface. They further increased the tumor specificity of the ADC by targeting gPD-L1 [82]. Furthermore, Li et al. proposed using gPD-1 mAbs to increase target specificity and reduce off-target effects of ADCs when the EGFR/B3GNT3/gPD-L1 axis is upregulated in TNBC cells [33]. In addition, combining ADCs with anti-PD-1/PD-L1 mAbs is a potential strategy to improve immunotherapy [83,84]. However, it has been shown that high PD-L1 expression on tumor-infiltrating lymphocytes is an independent favorable prognostic factor for surgically resected HNSCC [85]. The combination of ADCs with anti-PD-1/PD-L1 mAbs still needs further study and should be carefully evaluated in different tumors.

#### 4.4.3. Glycoengineered Therapeutic Antibodies

Glycoengineering techniques have the potential to optimize anti-PD-1/PD-L1 mAbs. The Fc-mediated effector functions of mAbs can be controlled by modulating Fc–FcγRs interactions by optimizing the *N*-glycan structure on the IgG Fc-domain [79]. For example, a glycosylated variant of an anti-PD-L1 antibody with core-fucosylated *N*-glycans in its Fc region (92%) led to an improvement in binding to FcγRIIIa, resulting in enhanced antibody-dependent cellular cytotoxicity (ADCC) activity against PD-L1^+^ cancer cells [86]. In addition to Fc region glycosylation, some antibodies are also glycosylated in the Fab region, which may be critical for their target recognition and half-life [79]. Moreover, it has been demonstrated that elevated levels of sialylation are linked to diminished activity in ADCC assays. This may be attributed to lower binding affinity [87]. Christian et al. found that the sialylation of the IgG Fc fragment is linked to adverse effects [50]. These studies demonstrate the potential of glyco-engineered technology for optimizing antibodies.

#### 4.4.4. Lysosome-Targeting Chimeras (LYTACs)

A LYTAC is a heterobifunctional macromolecule comprising a target-binding macromolecule (e.g., mAb) that binds to the target protein (POI) and a molecule that binds to cell surface lysosome-targeting receptors (LTRs) (Figure 2C). Subsequently, the complex undergoes endocytosis, resulting in the degradation of extracellular and membrane-bound proteins through the lysosome [88,89,90]. In 2020, the Bertozzi group constructed a LYTAC for the first time. Ab-3 demonstrated a notable reduction in PD-L1 levels on the cell surface compared to treatment with unfunctionalized anti-PD-L1 or anti-PD-L1-poly(Gal*N*Ac) [88]. Li et al. devised a covalent aptamer–LYTAC that is capable of effectively degrading PD-L1 in vivo. Furthermore, it was demonstrated that LYTAC-induced PD-L1 degradation directly results in immunogenic apoptosis of tumor cells, triggering tumor-specific immune responses [91]. Su et al. designed a genetically modified chimera and fused it to the surface of bacterial outer membrane vesicles (OMVs), combining LYTAC-mediated inhibition of the PD-1/PD-L1 pathway with the immune activation of OMVs [92].

Furthermore, Pance et al. developed a cytokine-receptor-targeted chimera designated as KineTAC. They produced KineTAC carrying the interleukin-2 cytokine and targeting PD-1 to activate T cells, notably reducing the level of PD-1 on the cell surface. Produced without complex synthesis or bio-coupling, KineTAC is a versatile, modular targeted degradation platform capable of robust lysosomal degradation in a variety of cell types. [93]. In a separate study, the lysosome-targeted co-assemblies (LYTACAs) developed by Wang et al. induced lysosomal degradation of extracellular protein IL-17A and membrane protein PD-L1 in several scavenger receptor-A-expressing cell lines and demonstrated the potential to degrade proteins associated with asialoglycoprotein receptors [94]. In addition, Zhang et al. generated PD-L1-degrading iLYTACs with recombinant proteins fused to the insulin-like growth factor 2 gene and PD-L1 nanoantibodies, which were tested in MDA-MB-231 breast cancer cells. Endogenous PD-L1 levels were significantly attenuated after 48 hours of treatment with 200 nM prepared iLYTACs (to half) [95].

### 4.5. As Predictive Biomarkers

The mean ORR for PD-1/PD-L1 inhibitors was 19.56%, and the clinical benefit to patients was not sustained [96]. There is an urgent need for predictive biomarkers, as they help identify patients who can benefit from anti-PD-1/PD-L1 therapy, predict prognosis, and promote personalized treatment plans. The FDA has approved several biomarkers for predicting the efficacy of ICIs. These include the expression of PD-L1, the tumor mutational burden (TMB), and DNA repair defects, including defective mismatch repair (dMMR) and microsatellite instability-high (MSI-H) [97]. Some of the most commonly used serologic biomarkers in clinical practice for cancer diagnosis and the monitoring of malignant progression, as well as prognostic biomarkers of disease recurrence, are glycoproteins [21]. However, their application is constrained by relatively low specificity, which impedes their screening and diagnosis [21]. A promising approach to identifying more rational diagnostic and predictive biomarkers is to find new analytical methods or to discover new targets through glycobiology, for example, using glycomics to analyze specific glycoforms. A study used a quantitative MS-based approach to define the glyco-signatures of patients with advanced metastatic squamous cell carcinoma (SCC) or adenocarcinoma receiving anti-PD-1/PD-L1 therapy. The findings indicated that detecting site-specific glycoforms on serum IgG may serve as a valuable ex vivo complementary test, facilitating accurate clinical management [98].

Glycoconjugates can serve as a source of non-invasive biomarkers. For example, a study demonstrated that T153, S157, S159, and T168 of alpha-fetoprotein (AFP) are modified with sialylated mucin-type O-glycans with core 1- and core 2-based structures [99]. Further meta-analysis showed that AFP-L3% could be complementary to AFP as a marker for HCC [100]. Additionally, the core fucosylation of alpha-fetoprotein has been approved as a biomarker for the early detection of HCC [99]. Apart from that, PD-L1 expression is a valuable biomarker for predicting the prognosis of and sensitivity to PD-1/PD-L1 blockade therapies. However, the correlation between PD-L1 immunohistochemistry readouts and patient responses is not straightforward, which presents a clinical challenge in patient stratification. Notably, PD-L1 deglycosylation has been shown to markedly enhance the binding affinity and signal intensity of anti-PD-L1 antibodies, improving the accuracy of PD-L1 detection. This PD-L1 antigen extraction method has the potential to reduce false-negative patients and facilitate patient stratification [101]. Deglycosylated PD-L1 has been a promising biomarker [41].

In comparison to non-transformed cells, tumor cells display extensive glycosylation alterations, which are primarily attributed to the aberrant expression and activity of GTs and glycosidases [21,102]. Consequently, these proteins and relative genes are potential biomarkers. For example, Lv et al. screened nine GT genes and established a glyco-signature for predicting the prognosis of patients with breast cancer (BC), demonstrating the potential of comprehensive bioinformatics analysis in prognostic assessment [103]. Xu et al. proposed that FUT7 overexpression is associated with poor prognosis [74]. In addition, the elevation of β-1,3-*N*-Glc*N*Ac transferase expression results in the aberrant glycosylation of PD-L1, which is linked to the onset of TNBC [26]. These have the potential to be markers. Furthermore, the predictive effect of a single biomarker is constrained, and a combined prediction model incorporating multiple biomarkers may be developed in the future [104,105,106].

## 5. Conclusions

Cancer cells exhibit immune evasion and immunosuppression, which challenge tumor immunotherapy. Despite the advent of ICIs, the treatment remains challenging. Unsatisfactory response rates, immune-related adverse events, drug resistance, and difficulties in accurate patient stratification and diagnosis have hindered these approaches. In this review, we highlighted the role of glycosylation in cancer progression and its potential to be a therapeutic target. We also explored the potential of glycobiological techniques to optimize PD-1/PD-L1 blockade therapies. Firstly, the extensive glycosylation of PD-1/PD-L1 impacts their structures and biological functions. Targeting glycoconjugates (e.g., gPD-L1) and their biosynthetic processes is promising. In addition, recent developments in glycoscience can help identify novel targets or improve therapeutic antibodies. With glycoscience, people have found new predictive biomarkers and improved detection of glycoproteins, facilitating accurate stratification and prognostication. For example, deglycosylation of PD-L1 may promote more precise detection in clinical settings. These avenues represent the potential of the emerging field of glycomedicine. Additionally, ADCs have the potential to overcome resistance to PD-1/PD-L1 blockade therapies and expand the potential beneficiary patients. ADCs may have broad applications in the future. In brief, advances in glycobiology have deepened the understanding of the relationship between glycosylation and cancer progression, optimized detection accuracy, and improved the clinical outcomes of therapeutic antibodies [33]. Nevertheless, PD-1/PD-L1 glycosylation and its role in tumor progression remain to be explored. A series of clinical trials are needed to verify whether patients in the real world can benefit from these strategies. The challenge of targeting glycosylation is to improve specificity and reduce side effects.

## Figures and Tables

**Figure 1 ijms-26-01238-f001:**
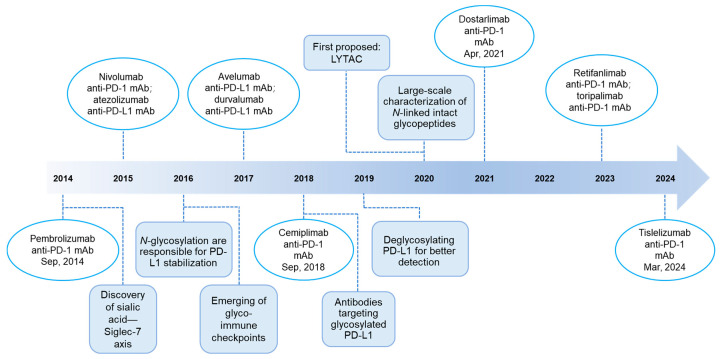
Timeline of FDA approval of PD-1/PD-L1-blocking antibodies and key milestones in glycoscience that advanced the efficacy of PD-1/PD-L1-blocking therapies. Abbreviations: mAb, monoclonal antibody; Siglec, sialic-acid-binding immunoglobulin (Ig)-like lectin; LYTAC, lysosome-targeting chimera.

**Figure 2 ijms-26-01238-f002:**
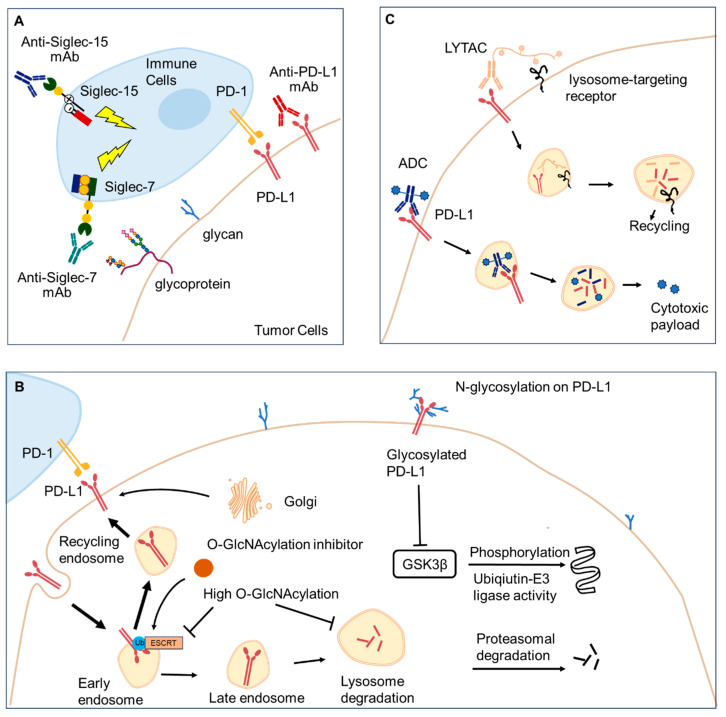
Representative strategies to improve PD-1/PD-L1 blockade therapies. (**A**) Targeting glyco-immune checkpoints. (**B**) *O*-Glc*N*Acylation inhibits the lysosome degradation of PD-L1. *N*-glycosylation on PD-L1 inhibits its proteasomal degradation. (C) Designing ADCs and LYTACs to block the PD-1/PD-L1 axis. Abbreviations: Siglec, sialic-acid-binding immunoglobulin (Ig)-like lectin; mAb, monoclonal antibody; GSK3β, glycogen synthase kinase 3β; ESCRT, endosomal sorting complexes required for transport; ADC, antibody–drug conjugate; LYTAC, lysosome-targeting chimera.

**Table 1 ijms-26-01238-t001:** Summary of current anti-PD-1 and anti-PD-L1 antibodies approved by FDA.

Target	Generic Name	Trade Name	Company	Approved
PD-1	Nivolumab	Opdivo, BMS-936558, MDX1106	Bristol-Meyers Squibb	Melanoma; NSCLC
	Pembrolizumab	Keytruda, MK-3475, Lambrolizumab	Merck	Melanoma; NSCLC; MPM; HNSCC; cHL; PMBCL; Urothelial Cancer; dMMR CRC; Gastric Cancer; Esophageal Cancer; Cervical Cancer; HCC; BTC; MCC; RCC; EC; TMB-H Cancer; CSCC; TNBC
	Cemiplimab	Libtayo, REGN2810	Sanofi	CSCC; BCC; NSCLC
	Dostarlimab	Jemperili, TSR-042	GlaxoSmithKline	EC; dMMR recurrent or advanced Solid Tumors
	Retifanlimab	Zynyz, INCMGA00012	Incyte	MCC
	Toripalimab	Loqtorzi, sintilimab	Junshi Biosciences	NPC
	Tislelizumab	Tevimbra, BGB-A317	BeiGene	ESCC
PD-L1	Atezolizumab	Tecentriq, MPDL3280A	Roche	NSCLC; SCLC; HCC; Melanoma; ASPS
	Avelumab	Bavencio, MSB0010718C	Merck, Pfizer	MCC; UC; RCC
	Durvalumab	Imfinzi, MEDI4736	AstraZeneca	NSCLC; ES-SCLC; BTC; uHCC; dMMR EC

**Abbreviations:** NSCLC: Non-Small Cell Lung Cancer; MPM: Malignant Pleural Mesothelioma; HNSCC: Head and Neck Squamous Cell Cancer; cHL: Classical Hodgkin Lymphoma; PMBCL: Primary Mediastinal Large B-Cell Lymphoma; dMMR: Microsatellite Instability-High or Mismatch Rrepair Deficient; CRC: Colorectal Cancer; HCC: Hepatocellular Carcinoma; BTC: Biliary Tract Cancer; MCC: Merkel cell carcinoma; RCC: Renal Cell Carcinoma; EC: endometrial cancer; TMB-H: Tumor Mutational Burden-High; CSCC: Cutaneous Squamous Cell Carcinoma; TNBC: Triple-Negative Breast Cancer; BCC: Basal Cell Carcinoma; NPC: nasopharyngeal carcinoma; ESCC: Esophageal Squamous Cell Carcinoma; SCLC: Small Cell Lung Cancer; ASPS: Alveolar Soft Part Sarcoma; UC: Urothelial Carcinoma; ES: Extensive-Stage; BTC: Biliary Tract Cancer; uHCC: unresectable HCC.

## Data Availability

No data were used for the research described in the article.

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
