# Peer review of "Glycoscience in Advancing PD-1/PD-L1-Axis-Targeted Tumor Immunotherapy"

_ijms, 2025, doi:10.3390/ijms26031238_

Round 1
Reviewer 1 Report
Comments and Suggestions for Authors
The topic addressed is complex, so I invite authors to simplify as better as possible, to make the lecture more fluid and to avoid repetitions.
The sentence “Concurrently, although patients respond to ICIs initially, disease progression, i.e., acquired resistance, may subsequently occurs” This sentence is generic since ICIs activity is tumor specific, for instance in metastatic melanoma at 10 yr the OS is 45% with anti PD1 and 50 % with IO combination, in desmoplastic melanoma is 70% (high TMB) as in MSI tumors.
Pag 2 “PD-1/PD-L1 blockade therapy is expected to become an important immunotherapeutic approach for various malignancies.” I think we could say that, nowadays, ICIs are an important immunotherapeutic approach, alone or combined with chemotherapy, they improved that clinical outcome in many solid tumors and in different settings (neoadjuvant, adjuvant and metastatic) as summarized in table 1
Pag 6 fig 2 I suggest explaining the cartoon steps more extensively for a better comprehension of the different mechanisms involved.
Pag 7 “Clinical trials are underway to block the Gal-3-LAG3 or Gal-9-TIM3 axis in conjunction with anti-PD-1 therapy. For further information, please refer to: …………” explain
Clinical data for LAG 3 inhibitors showed a very limited efficacy when used alone and when combined with anti PD1 an activity not superior to IO, even if the tolerance is better, long-term data are awaited. For TIM 3 inhibitors preclinical studies show promising results, however the expression of TIM-3 on various immune cells means that inhibitors might overstimulate the immune system, leading to immune-related adverse effects. The ultimate efficacy and safety of TIM-3 small molecule inhibitors must be determined through adequate clinical trials. Comment on 10.1016/j.ejmech.2024.117141.
“It is important to acknowledge that the selectivity, therapeutic efficacy, and oral bioavailability of targeting glycosylation still require further optimization”
Page 8. “Compared to monotherapy, the combination of OSMI-4 with anti-PD-L1 mAb demonstrated a synergistic inhibitory effect on the growth of hepatocellular carcinoma (HCC) and melanoma” specify in the preclinical setting.
Pag 8: PDL1 could be expressed also on TIL and a high PD-L1 expression on TILs is an independent favorable prognostic factor for surgically resected HNSCC. (see 10.1038/srep36956) therefore the use of ADC with anti PD1 should be carefully evaluated in different tumors.
References 99 precise these data were preclinical
Pag 10: “The response rate of PD-1/PD-L1 inhibitors is low and inconsistent among patients.” I found this statement a bit strong and scientifically untrue. The clinical effect of PD1/PDL1 inhibitors is greater than the response rate and stable diseases lasting > 6 months translate in an increased survival. This must be acknowledged, as well as the necessity of a long follow-up to obtain reliable data on new combinations or inhibitors.
Pag 10: “the tumor mutational burden (TMB), and DNA repair defects, including defective mismatch repair (dMMR) and microsatellite instability-high (MSI-H)……. are constrained by relatively low specificity, which impedes their screening and diagnosis”. Clinical data do not support this statement, and I completely disagree with it.
Pag 11: “Nevertheless, it is important to acknowledge that the precise alterations in PD-1/PD-L1 N-glycosylation linked to cancer development remain unclear, and further efforts are required.”
I think this point is crucial, and before verifying these strategies in real world patients, we have to prove them in clinical adequate studies.
Author Response
The topic addressed is complex, so I invite authors to simplify as better as possible, to make the lecture more fluid and to avoid repetitions.
R: We removed the irrelevant parts, which include “Notably, the N-acetylglucosamine bifurcation on N-glycans inhibits the glycosylation of various terminal epitopes in N-glycans, including Le-type fucose and sialic acid. This ultimately changes the overall conformation of the N-glycan [40]. The majority of cell surface proteins, including growth factor receptors, undergo N-glycosylation. Bisecting GlcNAc has been demonstrated to impede the epithelial-to-mesenchymal transition (EMT) triggered by transforming growth factor β receptor 1 (TGF-β1) [41]. This may be because bisecting Glc-NAc inhibits the interaction of N-acetyllactosamine with galectin-3 (Gal-3), which enhances TGF-β signaling by inhibiting the endocytosis of TGF-β receptors [42, 43].” “In addition, metformin has been demonstrated to promote the aberrant glycosylation of PD-L1 [74]. While targeting PD-L1 by regulating glycosylation offers promising prospects for a range of applications, it is important to acknowledge that the selectivity, therapeutic efficacy, and oral bioavailability of targeting glycosylation still require further optimization [10, 75]. Moreover, the interaction between RSV and PD-L1 is complex, which may restrict the applications of RSV [75].”
The sentence “Concurrently, although patients respond to ICIs initially, disease progression, i.e., acquired resistance, may subsequently occurs” This sentence is generic since ICIs activity is tumor specific, for instance in metastatic melanoma at 10 yr the OS is 45% with anti PD1 and 50 % with IO combination, in desmoplastic melanoma is 70% (high TMB) as in MSI tumors.
R: Thanks. We revised the text with a narrative describing the low response rate to PD-1/PD-L1 blockade therapy in certain cancers, as “Concurrently, heterogeneous responses have been seen even between distinct tumors within the same patient. Besides, patients with breast, prostate, and colon cancers have shown very low frequency of response, which has significantly reduced the clinical benefit [1].”
Pag 2 “PD-1/PD-L1 blockade therapy is expected to become an important immunotherapeutic approach for various malignancies.” I think we could say that, nowadays, ICIs are an important immunotherapeutic approach, alone or combined with chemotherapy, they improved that clinical outcome in many solid tumors and in different settings (neoadjuvant, adjuvant and metastatic) as summarized in table 1
R: Thanks a lot for these comments and we revised the manuscript accordingly. We modified the subject of this sentence to PD-1/PD-L1 blockades and took your narrative.
Pag 6 fig 2 I suggest explaining the cartoon steps more extensively for a better comprehension of the different mechanisms involved.
R: Thanks. We have revised Figure 2 and the annotation to present the content of the article.
Pag 7 “Clinical trials are underway to block the Gal-3-LAG3 or Gal-9-TIM3 axis in conjunction with anti-PD-1 therapy. For further information, please refer to: …………” explain
Clinical data for LAG 3 inhibitors showed a very limited efficacy when used alone and when combined with anti PD1 an activity not superior to IO, even if the tolerance is better, long-term data are awaited. For TIM 3 inhibitors preclinical studies show promising results, however the expression of TIM-3 on various immune cells means that inhibitors might overstimulate the immune system, leading to immune-related adverse effects. The ultimate efficacy and safety of TIM-3 small molecule inhibitors must be determined through adequate clinical trials. Comment on 10.1016/j.ejmech.2024.117141.
R: We agree with reviewer, and the text was revised as “Clinical trials are underway to block the Gal-3-LAG3 or Gal-9-TIM3 axis in conjunction with anti-PD-1 therapy. In 2022, relatlimab (an anti-LAG-3 antibody) in combination with nivolumab was approved by the FDA and the European Medicines Agency (EMA) for treating adult and pediatric patients with unresectable or metastatic melanoma. Bispecific antibodies targeting PD-1 and LAG-3 have also been developed and are under clinical investigation in solid tumors [65]. The combination of anti-LAG-3 mAbs and pembrolizumab was also assessed in patients with metastatic NSCLC and head and neck cancer (NCT03625323). However, results from an open-label phase II study (NCT03662659) showed that in patients with gastric or gastroesophageal adenocarcinoma, the objective response rate (ORR) was reduced in patients receiving the combination regimen of relatlimab plus nivolumab and chemotherapy versus nivolumab plus chemotherapy [66]. Besides the ongoing trials, further clinical studies on TIM-3 inhibitors in combination with anti-PD-1/PD-L1 mAbs are needed.”
“It is important to acknowledge that the selectivity, therapeutic efficacy, and oral bioavailability of targeting glycosylation still require further optimization”
R: The text was revised accordingly as “However, the oral bioavailability of RSV is considerably less than 1% [70].”
Page 8. “Compared to monotherapy, the combination of OSMI-4 with anti-PD-L1 mAb demonstrated a synergistic inhibitory effect on the growth of hepatocellular carcinoma (HCC) and melanoma” specify in the preclinical setting.
R: In this revision, the mentioned studies were discussed, as “Zhu et al. chemically synthesized CPP-G1 peptide containing HGS glycosylation sites. Treating SK-Hep-1 cells with different concentrations of CPP-G1 peptide decreased the expression of HGS O-GlcNAcylation and PD-L1 in a dose-dependent manner, but had no significant effect on the cellular O-GlcNAcylation level. ”
Pag 8: PDL1 could be expressed also on TIL and a high PD-L1 expression on TILs is an independent favorable prognostic factor for surgically resected HNSCC. (see 10.1038/srep36956) therefore the use of ADC with anti PD1 should be carefully evaluated in different tumors.
R: Thanks for the additions! We revised the description as following: “However, it has been shown that high PD-L1 expression on tumor-infiltrating lymphocytes is an independent favorable prognostic factor for surgically resected HNSCC [85]. The combination of ADCs with anti-PD-1/PD-L1 mAbs still needs further study and should be carefully evaluated in different tumors.”
References 99 precise these data were preclinical
R: We revised the statement as: “Besides, Zhang et al. generated PD-L1-degrading iLYTACs with recombinant proteins fused to the insulin-like growth factor 2 gene and PD-L1 nanoantibodies, which were tested in MDA-MB-231 breast cancer cells. Endogenous PD-L1 levels were significantly attenuated after 48 hours of treatment with 200 nM prepared iLYTAC (to half) [95].”
Pag 10: “The response rate of PD-1/PD-L1 inhibitors is low and inconsistent among patients.” I found this statement a bit strong and scientifically untrue. The clinical effect of PD1/PDL1 inhibitors is greater than the response rate and stable diseases lasting > 6 months translate in an increased survival. This must be acknowledged, as well as the necessity of a long follow-up to obtain reliable data on new combinations or inhibitors.
R: Sorry for this confusion. We revised the statement as: “The mean ORR for PD-1/PD-L1 inhibitors was 19.56% and the clinical benefit to patients was not sustained [96].”
Pag 10: “the tumor mutational burden (TMB), and DNA repair defects, including defective mismatch repair (dMMR) and microsatellite instability-high (MSI-H)……. are constrained by relatively low specificity, which impedes their screening and diagnosis”. Clinical data do not support this statement, and I completely disagree with it.
R: Sorry for this confusion. The subject of the second sentence is glycoprotein, so we add a statement before this sentence: “Besides, Some of the most commonly used serologic biomarkers in clinical practice for cancer diagnosis and monitoring of malignant progression, as well as prognostic biomarkers of disease recurrence, are glycoproteins [21].”
Pag 11: “Nevertheless, it is important to acknowledge that the precise alterations in PD-1/PD-L1 N-glycosylation linked to cancer development remain unclear, and further efforts are required.”
R: We revised the statement as: “Nevertheless, PD-1/PD-L1 glycosylation and its role in tumor progression remains to be explored.”
I think this point is crucial, and before verifying these strategies in real world patients, we have to prove them in clinical adequate studies.
R: Couldn't agree more. Thanks for pointing this out. We've refined the detailed descriptions in the article.
Reviewer 2 Report
Comments and Suggestions for Authors
It was a genuine pleasure to read the manuscript "Glycoscience in Advancing the PD-1/PD-L1 Axis Targeted Tumor Immunotherapy" by Qiyue Sun and Senlian Hong. The paper is well-written, featuring clear and precise English. The authors have managed to comprehensively cover a broad spectrum of topics related to the rather niche subject of PD-1/PD-L1 glycosylation.
The manuscript provides an excellent overview of glycosylation, detailing its various types and highlighting its significant role in cancer. This foundational knowledge sets the stage for understanding the subsequent sections. I found the section on the types and importance of glycosylation particularly well-done, although I would recommend including a figure to illustrate the modifications discussed in section 2.2. to help illustrate how complex biochemical processes are involved.
One of the standout sections of the manuscript is Chapter 3, which focuses on targeting glycosylation. The authors have outlined various attempts in this area, making it a very interesting read. However, it would be beneficial to see the strongest compounds referenced in this chapter presented as a separate figure preferably including their affinity to target proteins or IC50 values.
Additionally, the discussion on monoclonal antibodies targeting glycosylated PD-1/PD-L1, including LYTACS, is both fascinating and niche. This topic is of great interest and is well-addressed by the authors.
Overall, I fully support the publication of this manuscript. It is a significant contribution to the field of glycoscience and its application in tumor immunotherapy, providing valuable insights and a thorough exploration of the subject matter.
Author Response
It was a genuine pleasure to read the manuscript "Glycoscience in Advancing the PD-1/PD-L1 Axis Targeted Tumor Immunotherapy" by Qiyue Sun and Senlian Hong. The paper is well-written, featuring clear and precise English. The authors have managed to comprehensively cover a broad spectrum of topics related to the rather niche subject of PD-1/PD-L1 glycosylation.
The manuscript provides an excellent overview of glycosylation, detailing its various types and highlighting its significant role in cancer. This foundational knowledge sets the stage for understanding the subsequent sections. I found the section on the types and importance of glycosylation particularly well-done, although I would recommend including a figure to illustrate the modifications discussed in section 2.2. to help illustrate how complex biochemical processes are involved.
One of the standout sections of the manuscript is Chapter 3, which focuses on targeting glycosylation. The authors have outlined various attempts in this area, making it a very interesting read. However, it would be beneficial to see the strongest compounds referenced in this chapter presented as a separate figure preferably including their affinity to target proteins or IC50 values.
R: Thanks. We supplemented the IC50 information within the text. In addition to this, we also revised Figure 2.
Additionally, the discussion on monoclonal antibodies targeting glycosylated PD-1/PD-L1, including LYTACS, is both fascinating and niche. This topic is of great interest and is well-addressed by the authors.
Overall, I fully support the publication of this manuscript. It is a significant contribution to the field of glycoscience and its application in tumor immunotherapy, providing valuable insights and a thorough exploration of the subject matter.
Reviewer 3 Report
Comments and Suggestions for Authors
The manuscript titled "Glycoscience in Advancing the PD-1/PD-L1 Axis Targeted Tumor Immunotherapy" examines the role of glycosylation and related mechanisms in enhancing immune checkpoint blockade therapies. This report provides feedback on areas that could benefit from refinement, to enhance the presentation, analytical depth, and overall coherence.
Abstract
- The abstract is well-structured but lacks specific quantitative data or references to substantiate the claims, such as the impact of glycosylation on PD-1/PD-L1 binding.
- The language used is somewhat generic; it could benefit from more precise definitions or examples of how glycosylation optimizes PD-1/PD-L1 blockade therapy.
- Key terms such as "glycosylation" and "immune invasion" are not explained, assuming prior knowledge without providing context.
Figure 1: The resolution of Figure 1 is quite low.
Figure 2: The figure represents a very basic schematic rather than a polished scientific illustration. The drawing lacks uniformity in the representation of cellular components (e.g., endosomes, Golgi apparatus). Some structures appear hastily sketched compared to others, detracting from the overall professional appearance.
Section 1: PD-1/PD-L1 Axis
- The discussion on monoclonal antibodies is not sufficiently linked to the role of glycosylation, missing an opportunity to tie this section to the manuscript's central theme.
- The section introduces many molecular terms without cross-referencing to later discussions on glycosylation, leading to a fragmented narrative.
Section 2: Glycosylation
- Key claims regarding glycosylation as a druggable target and its role as a biomarker lack quantitative evidence or examples from the literature.
- The description of glycosylation changes in cancer progression is comprehensive but does not sufficiently emphasize how these alterations specifically affect the PD-1/PD-L1 axis.
Subsection 2.1: N-Glycosylation and O-Glycosylation
- The explanation of N-glycosylation processes is detailed, but the connection to the manuscript's focus on PD-1/PD-L1 immunotherapy is indirect.
- The description of PD-L1 glycosylation provides new insights but lacks clarity in linking glycosylation patterns to therapeutic implications.
- Some statements are redundant, such as those about PD-1 glycosylation sites, which are overly detailed and could be summarized for conciseness.
Subsection 3.2. Targeting Glycosylation
- Selective Targeting Challenges: While the section mentions challenges such as selectivity and bioavailability, it lacks depth in discussing these issues or strategies to address them.
- Insufficient Mechanistic Insight: Descriptions of how compounds such as D-mannose and resveratrol induce glycosylation changes are superficial, with limited mechanistic details provided.
Subsection 3.3. Targeting Glycoside Hydrolases and Glycosyltransferases (GTs)
Side Effects Minimally Addressed: The adverse effects of targeting glycosyltransferases are mentioned but not explored, which leaves gaps in understanding the therapeutic trade-offs.
Subsection 3.4. Optimizing PD-1/PD-L1 Blocking Therapeutic Antibodies
- Duplication in Subsections: The discussion of glycosylation-specific monoclonal antibodies (e.g., STM108) is repeated across different parts of the manuscript, which creates redundancy.
- Insufficient Validation for ADCs: The section does not discuss how the limitations of ADCs (e.g., toxicity, resistance) are addressed in ongoing research.
Subsection 3.4.4. Lysosome-Targeting Chimeras (LYTAC)
Insufficient Context on KineTAC: The discussion on KineTAC lacks clarity about its comparative advantages over other LYTAC approaches.
References: I would suggest considering the inclusion of more recent studies to ensure the literature review reflects the current state of research in this field.
These issues collectively reduce the clarity, depth, and impact of the manuscript. Addressing them would significantly enhance the review article’s quality and readability.
Author Response
The manuscript titled "Glycoscience in Advancing the PD-1/PD-L1 Axis Targeted Tumor Immunotherapy" examines the role of glycosylation and related mechanisms in enhancing immune checkpoint blockade therapies. This report provides feedback on areas that could benefit from refinement, to enhance the presentation, analytical depth, and overall coherence.
Abstract
- The abstract is well-structured but lacks specific quantitative data or references to substantiate the claims, such as the impact of glycosylation on PD-1/PD-L1 binding.
- The language used is somewhat generic; it could benefit from more precise definitions or examples of how glycosylation optimizes PD-1/PD-L1 blockade therapy.
- Key terms such as "glycosylation" and "immune invasion" are not explained, assuming prior knowledge without providing context.
R: Examples of optimizing PD-1/PD-L1 blockade by glycosylation were added to the abstract. “Recent achievements in glycoscience to enhance patient outcomes, referred to as glycotherapy, have underscored their high potency in advancing PD-1/PD-L1 blockade therapies, i.e., glycoengineered antibodies with improved binding towards PD-1/PD-L1, pharmaceutic inhibitors for core-fucosylation and sialylation, and synergistic treatment with the antibody-sialidase conjugate.” Besides, we have added an explanation of glycosylation in the abstract. “Current studies have established that glycosylation, a common post-translational modification, is crucial in promoting cancer progression and immune invasion.”
Figure 1: The resolution of Figure 1 is quite low.
R: Sorry for this. The image has been revised.
Figure 2: The figure represents a very basic schematic rather than a polished scientific illustration. The drawing lacks uniformity in the representation of cellular components (e.g., endosomes, Golgi apparatus). Some structures appear hastily sketched compared to others, detracting from the overall professional appearance.
R: We have revised Figure 2.
Section 1: PD-1/PD-L1 Axis
- The discussion on monoclonal antibodies is not sufficiently linked to the role of glycosylation, missing an opportunity to tie this section to the manuscript's central theme.
- The section introduces many molecular terms without cross-referencing to later discussions on glycosylation, leading to a fragmented narrative.
R:1. We revised the draft and included text as: “Nowadays, PD-1/PD-L1 blockades are popular, alone or combined with chemotherapy, to improve patient outcomes of many solid tumors in different settings (neoadjuvant, adjuvant, and metastatic) as summarized in Table 1.” This led to a stronger connection between the article and Table 1, but we failed to add elaboration on glycosylation.
- We added a description in subsection 2.2.2: “Siglecs function as inhibitory co-receptors and activating receptors for immune cell signaling in different contexts [54]. Upon binding of sialosides with inhibitory Siglecs, the intracellular domains of Siglecs can be phosphorylated by locally activated Src ki-nases, recruiting SHP-1 or SHP-2 phosphatases that decuple the downstream cellular activation [48].”
Section 2: Glycosylation
- Key claims regarding glycosylation as a druggable target and its role as a biomarker lack quantitative evidence or examples from the literature.
- The description of glycosylation changes in cancer progression is comprehensive but does not sufficiently emphasize how these alterations specifically affect the PD-1/PD-L1 axis.
R:1. We supplemented examples in section 3. In subsection 3.1: “Haas et al. demonstrated that targeting the tumor-restricted, glycosylation-dependent sialosides/Siglec-9 axis can release T cell subsets within the melanoma TME while simultaneously limiting T cell activation to the TME, reducing the potential for uncontrolled T-cell activation and immune-related adverse events [42].” “In 2022, relatlimab (an anti-LAG-3 antibody) in combination with nivolumab was approved by the FDA and the European Medicines Agency (EMA) for treating adult and pediatric patients with unresectable or metastatic melanoma. Bispecific antibodies targeting PD-1 and LAG-3 have also been developed and are under clinical investigation in solid tumors [65]. The combination of anti-LAG-3 mAbs and pembrolizumab was also assessed in patients with metastatic NSCLC and head and neck cancer (NCT03625323). However, results from an open-label phase II study (NCT03662659) showed that in patients with gastric or gastroesophageal adenocarcinoma, the objective response rate (ORR) was reduced in patients receiving the combination regimen of relatlimab plus nivolumab and chemotherapy versus nivolumab plus chemotherapy [66]. Besides the ongoing trials, further clinical studies on TIM-3 inhibitors in combination with anti-PD-1/PD-L1 mAbs are needed.” In subsection 3.2: “Besides, Zhang et al. found that niclosamide, an inhibitor acting on both HuR translocation and PD-L1 glycosylation, exhibited potent cytotoxicity against MDA-MB-231 with an half-maximal inhibitory concentrations (IC50) of 1.07 μM, owning potential to impair immunotherapy resistance[71].” In subsection 3.3: “Xu et al. postulated that fucosyltransferase VII (FUT7) remodels the glucose metabolism and TME of TNBC, becoming a potential target for enhancing ICB treatment [74].”
- We added sentences: “Besides, recent studies have demonstrated that PTMs of the PD-L1 control its stability and interaction with PD-1 [31]. Accumulating evidence has shown PD-1/PD-L1 is highly glycosylated [32, 33].”
Subsection 2.1: N-Glycosylation and O-Glycosylation
- The explanation of N-glycosylation processes is detailed, but the connection to the manuscript's focus on PD-1/PD-L1 immunotherapy is indirect.
- The description of PD-L1 glycosylation provides new insights but lacks clarity in linking glycosylation patterns to therapeutic implications.
- Some statements are redundant, such as those about PD-1 glycosylation sites, which are overly detailed and could be summarized for conciseness.
R:1. We removed some text: “Notably, the N-acetylglucosamine bifurcation on N-glycans inhibits the glycosylation of various terminal epitopes in N-glycans, including Le-type fucose and sialic acid. This ultimately changes the overall conformation of the N-glycan [40]. The majority of cell surface proteins, including growth factor receptors, undergo N-glycosylation. Bisecting GlcNAc has been demonstrated to impede the epithelial-to-mesenchymal transition (EMT) triggered by transforming growth factor β receptor 1 (TGF-β1) [41]. This may be because bisecting Glc-NAc inhibits the interaction of N-acetyllactosamine with galectin-3 (Gal-3), which enhances TGF-βsignaling by inhibiting the endocytosis of TGF-β receptors [42, 43].” We think this helphighlight the relationship between glycosylation and PD-1/PD-L1 axis.
- We explain this in section 3: “The loss of core fucosylation markedly enhances PD-1 ubiquitination, resulting in PD-1 degradation within the proteasome [56]. Interrupting core fucosylation offers a promising strategy to improve anti-tumor immune responses [45, 57].” “Haas et al. demonstrated that targeting the tumor-restricted, glycosylation-dependent sialosides/Siglec-9 axis can release T cell subsets within the melanoma TME while simultaneously limiting T cell activation to the TME, reducing the potential for uncontrolled T-cell activation and immune-related adverse events [42].”
- In subsection 3.4.1, we reviewed strategies for targeting glycosylated PD-1/PD-L1 antibodies: “Moreover, it has demonstrated that mAbs that recognize the gPD-L1 N192/N200 site, rather than those that specifically target N35, are the only mAbs that can exert induction effects.” “Apart from that, Wang et al. developed MW11-h317 (a mAb targeting gPD-1). The crystal structure revealed that the N58 glycosylation of PD-1 was a critical determinant in the binding process.” “Similarly, their findings revealed that N58 in the BC loop played a crucial role in mediating the interaction between mAb059c and PD-1.” We thought this part was necessary for readers to know PD-1 glycosylation sites first.
Subsection 3.2. Targeting Glycosylation
- Selective Targeting Challenges: While the section mentions challenges such as selectivity and bioavailability, it lacks depth in discussing these issues or strategies to address them.
- Insufficient Mechanistic Insight: Descriptions of how compounds such as D-mannose and resveratrol induce glycosylation changes are superficial, with limited mechanistic details provided.
R:1. We described specific data on the oral bioavailability of RSV: “However, the oral bioavailability of RSV is considerably less than 1% [70]. Moreover, the interaction between RSV and PD-L1 is complex [69]. These may restrict the applications of RSV.”
- We describe the mechanism in detail: “It has been demonstrated that D-mannose can activate AMP-activated protein kinase (AMPK) to phosphorylate PD-L1 at S195, leading to abnormal glycosylation and proteasomal degradation of PD-L1 (Figure 2B).” “Similarly, studies have shown that AMPK activated by metformin directly phosphorylates S195 of PD-L1 [68].”
Subsection 3.3. Targeting Glycoside Hydrolases and Glycosyltransferases (GTs)
Side Effects Minimally Addressed: The adverse effects of targeting glycosyltransferases are mentioned but not explored, which leaves gaps in understanding the therapeutic trade-offs.
R: We explored the OGT inhibitors limitations: “Reported OGT inhibitors include substrate analogs, dual-substrate inhibitors, and high-throughput screening inhibitors [75]. These OGT inhibitors have some drawbacks for their clinical applications, as shown by reducing HBP flux and jeopardizing glycan synthesis, poor cell permeability, poor selectivity, and off-target toxicity [76].”
Subsection 3.4. Optimizing PD-1/PD-L1 Blocking Therapeutic Antibodies
- Duplication in Subsections: The discussion of glycosylation-specific monoclonal antibodies (e.g., STM108) is repeated across different parts of the manuscript, which creates redundancy.
- Insufficient Validation for ADCs: The section does not discuss how the limitations of ADCs (e.g., toxicity, resistance) are addressed in ongoing research.
R:1. We've removed redundant narratives.
- We added text to highlight the role of glycoscience in optimizing ADCs as: “Besides, Li et al. proposed that using gPD-1 mAbs to increase target specificity and reduce off-target effects of ADCs when the EGFR/B3GNT3/gPD-L1 axis is upregulated in TNBC cells [33].”
Subsection 3.4.4. Lysosome-Targeting Chimeras (LYTAC)
Insufficient Context on KineTAC: The discussion on KineTAC lacks clarity about its comparative advantages over other LYTAC approaches.
R: We added a comparison for the advantages of KineTAC. “Produced without complex synthesis or biocoupling, KineTAC is a versatile, modular targeted degradation platform capable of robust lysosomal degradation in a variety of cell types. [93].”
References: I would suggest considering the inclusion of more recent studies to ensure the literature review reflects the current state of research in this field.
R: Thanks. We added a reference: “71. Zhang, Q.; Yang, Z.; Hao, X.; Dandreo, L. J.; He, L.; Zhang, Y.; Wang, F.; Wu, X.; Xu, L., Niclosamide improves cancer immunotherapy by modulating RNA-binding protein HuR-mediated PD-L1 signaling. Cell & Bioscience 2023, 13, (1).” We added this to fully demonstrate that targeting PD-L1 glycosylation owning potential to impair immunotherapy resistance.
These issues collectively reduce the clarity, depth, and impact of the manuscript. Addressing them would significantly enhance the review article’s quality and readability.
Reviewer 4 Report
Comments and Suggestions for Authors
The manuscript entitled “Glycoscience in advancing the PD-1/PD-L1 axis targeted tumor immunotherapy” submitted by Sun et.al. discusses the role of glycosylation in the context of PD-1/PD-L1 blockade therapy, highlighting the limitations of current immune checkpoint inhibitors (ICIs) and the potential to enhance their efficacy through an understanding of glycosylation patterns in tumor progression and immune response. The author provides an overview of the PD-1/PD-L1 pathway, its therapeutic significance, and outlines how glycosylation affects the immune response and immunotherapy outcomes. The article aims to offer new insights into how targeting glycosylation could optimize the efficacy of PD-1/PD-L1-based immunotherapies. This manuscript addresses an important and timely issue in cancer immunotherapy. While the topic holds significant potential for advancing the understanding of PD-1/PD-L1 blockade therapies, the manuscript would benefit from stronger data support, a more detailed discussion of immune-related adverse events, and greater clarity in the writing. The authors should include and explain how glycosylation may contribute to the modulation of irAEs or methods to reduce these side effects could add substantial value to the manuscript, as this is a critical aspect of immunotherapy.
Overall, the work done by Sun et.al is commendable.
Author Response
The manuscript entitled “Glycoscience in advancing the PD-1/PD-L1 axis targeted tumor immunotherapy” submitted by Sun et.al. discusses the role of glycosylation in the context of PD-1/PD-L1 blockade therapy, highlighting the limitations of current immune checkpoint inhibitors (ICIs) and the potential to enhance their efficacy through an understanding of glycosylation patterns in tumor progression and immune response. The author provides an overview of the PD-1/PD-L1 pathway, its therapeutic significance, and outlines how glycosylation affects the immune response and immunotherapy outcomes. The article aims to offer new insights into how targeting glycosylation could optimize the efficacy of PD-1/PD-L1-based immunotherapies. This manuscript addresses an important and timely issue in cancer immunotherapy. While the topic holds significant potential for advancing the understanding of PD-1/PD-L1 blockade therapies, the manuscript would benefit from stronger data support, a more detailed discussion of immune-related adverse events, and greater clarity in the writing. The authors should include and explain how glycosylation may contribute to the modulation of irAEs or methods to reduce these side effects could add substantial value to the manuscript, as this is a critical aspect of immunotherapy.
Overall, the work done by Sun et.al is commendable.
R: Thanks a lot for your suggestions. In this vision, we added some examples of glycoscience increasing antibody specificity, stated the IC50 of the compounds, and added the mechanism of PD-L1 degradation inhibition by N/O-glycosylation in section 3.
Round 2
Reviewer 3 Report
Comments and Suggestions for Authors
The recommended changes have been correctly incorporated into the revised version.